# Impact of Employee Job Attitudes on Ecological Green Behavior in Hospitality Sector

**Muhammad Arshad** [1,*] , **Ghulam Abid** [2] , **Jamil Ahmad** [1] , **Leena Anum** [3] **and Mumtaz Muhammad Khan** [4]

1 School of Business Administration, National College of Business Administration and Economics, Lahore 54660, Pakistan; Jamil_nbp@yahoo.com
2 Department of Business Administration, Kinnaird College for Women, Lahore 540000, Pakistan; dr.ghulamabid@gmail.com
3 Department of Management Sciences, Lahore Garrison University, Lahore 54810, Pakistan; leena.rehman@hotmail.com
4 Department Management Sciences, Imperial College of Business Studies, Lahore 53720, Pakistan; mumtazmkpk1@gmail.com
* Correspondence: arshad.tevta@gmail.com

**Abstract:** Notwithstanding the significant contribution made by employees towards addressing environmental issues, few research studies have explored this important contemporary theme in the hospitality sector. Drawing on the theory of planned behavior (TPB), this research examines the direct and interactive effects of employee environmental job attitudes and behaviors on ecological practices. Using PROCESS Macros on an actual convenient sample of 508 employees working in the hospitality industry, the results show a mixture of anticipated and surprising outcomes. The anticipated outcome is associated with the direct effects of environmental attitude on ecological behavior, while surprising outcomes are in the interaction of job attitudes and behavior (customer-oriented discretionary behavior, organizational commitment). These outcomes provoke employees' green behavior and contentment with the organization. The originality of this research is to investigate the significant contribution of employees in greening the hospitality sector in an emerging economy.

**Keywords:** environmental intent; customer-oriented OCB; organizational commitment; employee satisfaction; ecological behavior

## 1. Introduction

The hospitality sector is an important aspect of the service industry, sharing significant contributions to the world economy [1]. However, due to rapid industrialization in the recent years and global warming, this sector is facing serious environmental challenges [2]. Previous studies have noticed that the hospitality sector has demonstrated marginal emphasis on environmental issues, particularly on developing employee green behavior, which is crucial for meeting the customers' expectations [3]. It is because of the propagation of recent environmental legislations and mounting market pressure that hotels, restaurants and rest houses are realizing the importance of green practices for resolving environmental issues in the hospitality sector [4]. Keeping the importance of green practices in mind, hospitality organizations are training their employees to educate visitors and customers to reduce waste, and preserve resources such as energy and water in their daily usage [5]. Former studies were focused on green marketing with the view to address the demand and expectations of the customers [6]. Employees, being the active agents to actualize the vision of environmental issues, were less focused and, particularly, the role of employee job attitudes in the context of environmental management performance was overlooked and thus requires further exploration [3]. Employee environmental attitudes are important antecedents of ecological behavior which trigger the employee's intention to execute ecological practices. Furthermore, the role of job attitudes and the behavior of human

resource have been scarcely discussed. Previous studies also recommended that the effect of employee environmental commitment and discretionary behavior should be explored in resolving environmental issues [3,7]. Therefore, the results of this study could assist managers of the hospitality sector to design strategies and policies to acknowledge the significant contributions of employee commitment and citizenship behavior in greening the hospitality sector. While addressing the gap, this study adds to the theoretical literature of employee organizational commitment and citizenship behavior in highlighting the significant impact of employee environmental attitudes on job attitudes and ecological behavior and understanding what drives the employees to engage in labor intensive ecological behavior (EB) in greening the hospitality sector.

Extending the discussion further, to gain a sustainable competitive market advantage, business organizations use green practices to develop a soft brand image and meet the expectations of their customers [8]. Management approaches pertaining to environmental sustainability are vital for environmental performance and sustainable market advantage [9]. Furthermore, the implementation of green environmental practices in the hospitality sector adds value to the relevant HR by enriching their knowledge and skills [10]. This in turn provokes their green behavior and inclination to implement environmental practices in the organization [11]. Previous studies have investigated the impact of employee environmental attitude on EB [12], however, the indirect effect of employee job attitudes and behaviors between their relationships was not tested. To study the predictors of employee behavioral intentions, the theory of planned behavior (TPB) is explored. This theory argues that an employee's behavioral intention is mutually influenced by the attitude towards the behavior, individual norms, and perceived behavioral control [13]. TPB has often been approved by communal psychologists to envisage an employee's behavioral intention [14]. Anchored in the theory of planned behavior [13], this study aims to discuss the relationship between employee environmental attitudes and job attitudes. Additionally, the collaborating effects of the employee environmental and job attitudes on EB are investigated through an empirical approach, applied to the hotel industry located in an emerging, yet under-studied context: Pakistan. Therefore, this study attempts to bridge two important topics in tourism management: employee environmental attitude [3] and job attitudes [15], in the context of the hospitality sector.

## 2. Literature Review and Hypothesis Development

### 2.1. Theoretical Perspective

Employee participation is vital for the successful implementation of environmental strategies [3]. Acknowledging employees' participation is not merely essential for their satisfaction and well-being, but it is also a smart approach on the part of the organization for improving the working environment, open innovation, and creativity in the hospitality sector.

Open innovation is perceived as a fundamental factor for strategy maintenance that ensures affluence, sustainability, and the market advantage of any organization [16,17] and, most importantly, for the hospitality sector too. Open innovation imports novel values [16] and understanding from the external stakeholders which promotes in-house open innovation [18]. Knowledge conceived by external stakeholders usually spreads over several actors (chef, waiter, attendant, supervisor, and administrative staff) and it is often extended from an individual employee to the whole industry [19]. Empirical studies on innovation disclose that employees holding multiple skills and wide experiences, participate in effective innovation development [20]. Additionally, employees who contribute to innovation processes face the above mentioned challenges along with their household chores. However, it is significant to understand that the open innovation process is associated with a mindset stimulated by open innovation culture. This type of working environment is promoted through openness because employees with diversified knowledge and skills respond to the changing market [21], particularly for greening the hospitality sector. Therefore, executive management of the hospitality sector

encourages and acknowledges the contribution of its employees for the effective execution of environmental management systems.

This study explores the impact of employee job attitudes in developing employee EB for greening the hospitality sector. In the context of greening the hospitality sector, TPB assists in predicting the employee EB. Employees' intend to demonstrate a specific behavior effects the overall behavior and conduct. Therefore, for the execution of environmental proposals, the hotel staff is required to perform extra job activities for the protection of the environment and enhancement of organizational performance [22]. For instance, office staff may use double-sided printing or photocopying and housekeepers may be directed to adjust guest room temperatures and sort garbage for recyclable items, such as plastic bottles [23]. Cooks may be advised to turn on cooking tools when required, and not keep them on at the end of the shift. The literature suggests that around 15% of the entire electricity and fuel consumption in the hospitality industry is consumed by the kitchen [24]. Laundry staff is instructed to run full loads and cut linens into small bits for other purposes. Procurement staff may have to invest extra time to explore and procure products and tools which are environmentally friendly [25]. A planned environmental management system (EMS) may require better record keeping which requires the responsible supervisory staff to exert honest and extra effort to find ways to manage proper documentation. To address the less investigated gap, this study proposes important triggers, ranging from employee environmental intent (EI) to customer-oriented organizational citizenship behavior (OCB), organizational commitment (OC), and employee satisfaction (ES), that could encourage employee EB for the implementation of green practices in the hotel industry.

### 2.2. EI, Customer-Oriented OCB, OC and EB

Environmental intent (EI) refers to the willingness of the employee to use energy-saving or environmentally friendly materials with the aim of protecting the environment [23]. Previous studies have associated the concept of EI with environmental attitudes, i.e., environmental awareness, environmental knowledge, and environmental concern. They have recommended that employees' environmental attitudes should be congruent with the employee intent to implement environmental green practices [3]. Employee intent in the execution of green practices also relates to the work environment of the hotel firm, such as extra workload, the participation of supervisors and leadership, and communication systems that would influence employee EB [26]. An environmental management system (EMS) improves employees' EB when the organization is willing to implement environmental green practices [27]. Attitude of an individual employee serving in the hospitality sector plays a significant role in implementing environmentally green practices [4]. Environmental intent has largely been focused on environmental attitudes and its relationships with employee behaviors, i.e., citizenship behavior, organizational commitment (OC), and employee satisfaction (ES). Scholarly attention has been given to the individual pro-environmental behaviors in the hospitality sector as employees are agents to enact organizational green practices [28]. The paradigm shifts in research, from organizational green behavior to individual green behavior, assists in understanding the contribution of organizational citizenship behavior for environmental performance [29]. An employee engaged in environmental practices helps other colleagues to realize the significance of green practices for the sustainability of both the hospitality sector and the community [30].

However, an employee demonstrating customer-oriented discretionary behavior serves as a magnifier in predicting the employee EB in connection with employee intention for implementing green practices [29]. Still, there is scarce knowledge on the relationship between EI and EB. This literature guides us to further explore the relationship between employee extra-role behavior and EB. Researchers agree that employee pro-environmental behavior is associated with willingness to practice pro-environmental activities [31]. This discussion anticipates that the extra-role behavior promotes pro-environmental attitudes and could be an effective force for implementing green practices. Furthermore, engaging employees in addressing environmental issues is a useful strategy to be an environmentally

responsible organization and to improve environmental performance [31]. Discussion on the relationship between customer-oriented OCB and EB requires further deliberation, therefore, this study opted to empirically explore the affiliation between the employee customer-oriented OCB and EB.

A review of the association between OC and EB with respect to EI reveals that employees' EI is an important factor for implementing environmental management systems. Two dimensions of OC, continuance (determination to remain with the organization and not to leave due to potential loss of organization) and normative commitment (feeling an obligation to remain with the organization on ethical grounds) strengthen the employee relationship with the organization and stimulate their EB [32], OC, and its connection with several work domains and related outcomes, including employee participation [33], service quality [34], and turnover [35]. However, the impact of OC and its effect on employee environmental attitude (i.e., EB) in the hospitality sector requires further investigation.

Employees' EB and their impact on the natural environment are matters of public interest, and have been the subject of choice for scholars of psychological research [36]. Numerous studies have discussed the antecedents of individual EB, and also elucidate how these antecedents can be measured [37]. For instance, [38] referred to TPB to define the causes of EB that arise from behavior intention, looking at two factors: attitude towards behavior and subjective norms. Previous studies also support the notion that a connection to nature is linked with employees' pro-environmental behavior that provokes them to mitigate negative actions towards nature and promote green practices at workplace [39]. Scholars also noticed that both local and social (subjective) norms influence EB [40]. In this context, the impact of environmental attitudes on EB was also positive [3].

The adoption of green practices by the business organization provides a sustainable market advantage by significantly contributing to forming a positive brand image, and satisfying customers' demands and expectations [41]. Previous studies have investigated the impact of environmental green practices on environmental improvement and customers [42]. However, very few studies have examined the antecedents of employees' EB [12]. Exploring the existing gap in theoretical knowledge, this study empirically examines how environmental attitudes (EI) influence the execution of green practices, with the assistance of customer-oriented discretionary behavior and OC, which are expected to predict employee EB. The research recommends that green human resource practices in the hospitality sector raise organizational performance and influence EB [43]. Thus, the literature review suggests that customer-oriented OCB and OC signify the relationship between environmental attitudes and EB.

**Hypothesis 1 (H1).** *Customer-oriented OCB mediates the relationship between EI and EB.*

**Hypothesis 2 (H2).** *OC mediates the relationship between EI and EB.*

**Hypothesis 3 (H3).** *Customer-oriented OCB and OC indirectly mediate the relationship between EI and EB.*

### 2.3. ES, EI, Customer-Oriented OCB, and OC

The review of the previous literature associates employee satisfaction with employees' attitude and feelings towards the job, and other job-related aspects, such as interaction with colleagues, acknowledgment, benefits, and working environment and situation [44]. In the hospitality sector, employee satisfaction is identified as one of the key factors contributing towards organizational success [45]. Profit and growth of the organization are stimulated by customer loyalty which is won by customer satisfaction. However, the loyalty of the customer is characterized by the fulfillment of their expectations and satisfaction, earned by the quality services offered by the loyal and satisfied employees [46]. It is because of high labor intensity, constant human interactions, and reliance on other coworkers that consistency in the employee satisfaction of the hospitality sector has become a challenging

task in hotel management [47]. Several empirical studies have also connected employee satisfaction with organizational performance, and this relationship has been termed as the "Holy Grail" by industrial psychologists [48]. This view is recommended by social exchange theory that suggests that relationships are promoted by mutual trust [49], and trust is developed when an individual or group does something good for another and, in reciprocation, the beneficiary develops a sense of obligation which instigates the need to do something positive for the kind partner [50].

Employee EI, being a global attitude based on the principals of TPB, stimulates employee customer-oriented OCB to instigate feelings of contentment with the organizational initiatives in the hospitality sector. Previous literature has focused on the relationships of EI and employee satisfaction with environmental attitudes, such as environmental concern [3], environmental awareness [51], environmental knowledge [52], and employee EB [3]. However, the existing literature has discussed the direct relationships of EI and employee satisfaction less. Researchers agree that job attitudes promote employee intent, awareness, and commitment for the implementation of green practices, which shows that an employee holding EI demonstrates satisfaction with the organization [53].

Scholars have evinced that an environmental management system promotes an ethical climate and employees feel good about themselves. This motivates them to demonstrate effective commitment [54]. Social identity theory also explores the relationship between environmental management and OC: individuals pursue their social identity to improve self-esteem [55]. Consequently, the employees feel pride in being part of a socially responsible organization and demonstrate positive work behavior (i.e., OC). Debate on the relationship between OC and employee satisfaction proposes that employee EI is linked with OC and employee satisfaction. Affective and normative commitment has a significant positive influence on employee satisfaction [56]. In this perspective, social identity theory discusses the impact of an ethical environment on employee attitude, and subsequently proposes a theoretical justification for the relationships between environmental management systems and OC, and argues that a positive perception of social identity causes a positive impact on employee attitude and stimulates OC [57].

OC and ES are interlinked, and play a significant role in the execution of environmental green practices. The mediating roles of OC in human resource studies have been explored in different contexts, i.e., leadership and turnover intention [58], organizational support, and employee retention [59]. The collaborative effect of both constructs, customer-oriented OCB and OC, at an organizational level, has been investigated much less in previous studies [60]. A highly committed employee demonstrates volunteer discretionary behavior in the workplace [61] and, conversely, an employee engaged in OCB, representing volunteer devotion, conscientiousness, and job commitment, may demonstrate OC that includes actions pertaining to green behavior with the ultimate objective to draw the attention of customers [11]. Therefore, employee EI and customer-oriented discretionary behavior represent volunteer attitudes and behavior, with the perception to win the customer sustainability in the long run and also endeavor to explicitly support organizational objectives, such as showing willingness and approving behavior to outsiders and also accepting the organizational policies for new change. The understanding of environmental initiatives in the employees is significant for the effective implementation of environmental practices [52]. The indirect impact of customer-oriented OCB and OC has been addressed in depth, in the context of human resource management, however, scarce research studies exist deliberating the mediating roles of employee customer-oriented discretionary behavior and work attitude. This study empirically examines the interactive effects of customer-oriented OCB and organizational commitment in the relationship of EI with employee satisfaction.

**Hypothesis 4 (H4).** *Customer-oriented OCB mediates the relationship between EI and ES.*

**Hypothesis 5 (H5).** *OC mediates the relationship between EI and ES.*

**Hypothesis 6 (H6).** *Customer-oriented OCB and OC mediate the relationship between EI and ES.*

*2.4. Research Framework*

The framework presented in Figure 1 has been developed on the basis of a literature review in line with the concept of the theory of planned behavior. It represents the role of employee job attitudes and behaviors in the relationship of employee environmental attitude and EB within the background of greening the hospitality sector.

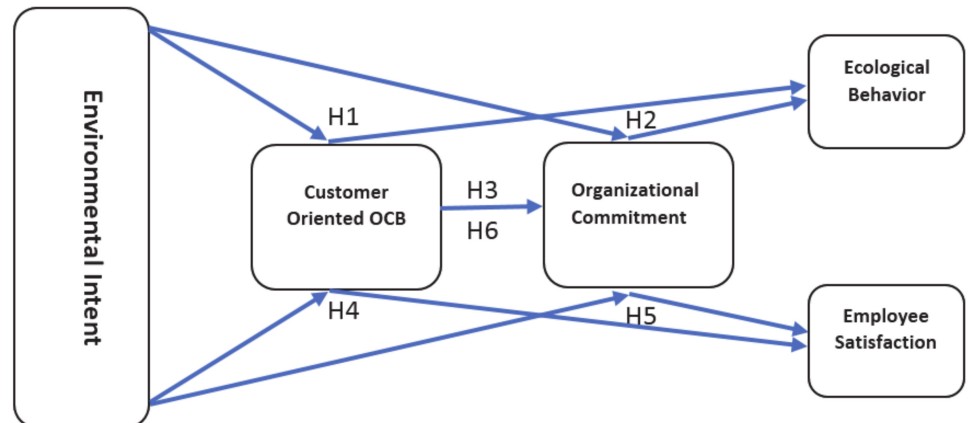

**Figure 1.** Research Framework.

## 3. Methodology

*Measures*

A self-administrative survey questionnaire was developed to measure the proposed model (Figure 1). All the constructs in the questionnaire were measured using pre-tested scales. The data were collected for this cross-sectional study from 508 managerial and supporting staff, selected on a convenient sampling basis, of the hotel industry, having at least one year of experience, through survey questionnaires T1 (demographic variables, employee EI, customer-oriented OCB and OC) and T2 (ES and EB) to avoid common method bias [62]. The units of analysis for this study were the individuals as the respondents of our study. Out of 700 survey questionnaires (T1) distributed in 70 organizations, 588 were returned and, after two weeks, T2 survey questionnaires were served and 545 survey questionnaires were received, giving a response rate of 78%. Out 545 survey questionnaires, 508 (response rate 72%) useable survey questionnaires were selected and recorded in SPSS 24 for data analysis. The survey questionnaires (T1 and T2) consisted of five-point Likert scale question items, ranging from 1 (strongly disagree) to 5 (strongly agree). The study variables, their references, sample items, and internal consistency reliability are presented in Table 1.

**Table 1.** Study variables, reliability, sample items, and references.

| S.# | Constructs | Items | Sample Items | Reliability | References |
|---|---|---|---|---|---|
| 1 | EI | 5 | I would be willing to follow the hotel instructions to perform the required environmental practice materials in their operations whenever possible | 0.83 | [63] |
| 2 | CUOOCB | 8 | To serve my customers, I volunteer for things that are not required | 0.76 | [64] |
| 3 | OC | 6 | I am proud to tell others that I am part of this organization | 0.81 | [65] |
| 4 | ES | 3 | All in all, I am satisfied with my job | 0.79 | [66] |
| 5 | EB | 7 | I reuse my shopping bags | 0.85 | [67] |

**Note:** EI: Environmental intent, CUOOCB: customer-oriented OCB, OC: Organizational commitment, ES: Employee satisfaction, EB: Ecological behavior.

## 4. Data Analysis

### 4.1. Measurement Validation

Prior to assessing convergent and discriminant validity through confirmatory factor analysis (CFA), model fit indices were evaluated for our measurement model and alternate models. Initially, a full five-factor measurement model was examined. We drew all the items of our five study variables in AMOS 24 and then permitted the items to correlate liberally with their respective factors. The results of our hypothesized five-factor model (EI, customer-oriented OCB, OC, ES, EB) plausibly showed a good fit (Table 2), as incremental fit index (IFI) = 0.96, normed fit index (NFI) = 0.97, relative fit index (RFI) = 0.94, Tucker–Lewis index (TLI) = 0.93, confirmatory fit index (CFI) = 0.98, and root mean square error of approximation (RMSEA) = 0.06. All of these indices fall into the satisfactory limit: TLI > 0.90 [68], IFI > 0.90, CFI > 0.90, and RMSEA < 0.08 [69].

**Table 2.** Confirmatory factor analysis.

| Models | IFI | NFI | RFI | CFI | TLI | RMSEA |
|---|---|---|---|---|---|---|
| Five-Factor Model | 0.96 | 0.97 | 0.94 | 0.98 | 0.93 | 0.06 |
| Four-Factor Model | 0.84 | 0.82 | 0.75 | 0.82 | 0.81 | 0.07 |
| Three-Factor Model | 0.80 | 0.75 | 0.71 | 0.74 | 0.72 | 0.07 |
| Two-Factor Model | 0.74 | 0.71 | 0.62 | 0.71 | 0.69 | 0.08 |
| One-Factor Model | 0.71 | 0.65 | 0.58 | 0.64 | 0.61 | |

Five-Factor Model: All the factors individually; Four-Factor Model: EI and customer-oriented OCB combine into one factor; Three-Factor Model: EI and OC combine into one factor, ES and EB combine into one factor; Two-Factor Model: EI, customer-oriented OCB, and ES combine into one factor, OC and EB combine into one factor; One-Factor Model: All the variables combine into one factor.

The full (five-factor) measurement model was also compared with the alternate nested models in order to find a best fit model for our data. The results show that the five-factor model presents the best fit for our data and none of the alternate nested models provided an acceptable model fit at $p < 0.001$. Thus, the results provided support for the idea that EI, customer-oriented OCB, OC, ES, and EB are distinct constructs (Table 2).

### 4.2. Correlation Matrix

The findings show that employee EI is positively associated with ES ($r = 0.51$, $p < 0.001$) and EB ($r = 0.57$, $p < 0.01$). The findings explain that EI has a positive effect on customer-oriented OCB ($r = 0.50$, $p < 0.01$) and OC ($r = 0.56$, $p < 0.01$). Overall, the results also explore significant effects of education on employee customer-oriented OCB ($r = 0.09$, $p < 0.05$), OC ($r = 0.09$, $p < 0.05$), and EB ($r = 0.13$, $p < 0.01$), which indicate that highly educated employees demonstrate more responsibility and commitment towards organization and the implementation of green practices. The results further explore significant negative effects of job type on ES ($r = -0.12$, $p < 0.01$), which indicate that supporting staff were not satisfied with the organization (Table 3).

**Table 3.** Means, standard deviations, and correlations of the study variables.

| Variables | Mean | SD | 1 | 2 | 3 | 4 | 5 | 6 | 7 | 8 | 9 | 10 | 11 |
|---|---|---|---|---|---|---|---|---|---|---|---|---|---|
| 1. Gender | 1.22 | 0.42 | - | | | | | | | | | | |
| 2. Age | 2.30 | 0.81 | −0.04 | - | | | | | | | | | |
| 3. Education | 2.31 | 1.08 | 0.30 ** | 0.32 ** | - | | | | | | | | |
| 4. Job Type | 1.77 | 0.42 | 0.17 ** | −0.30 ** | −0.28 ** | - | | | | | | | |
| 5. Tenure | 1.62 | 0.86 | −0.22 | 0.48 ** | 0.09 | −0.36 ** | - | | | | | | |
| 6. Marital Status | 1.53 | 0.50 | −0.11 * | 0.37 ** | 0.13 ** | −0.12 ** | 0.25 ** | - | | | | | |
| 7. EI | 4.23 | 0.52 | −0.02 | 0.04 | 0.04 | −0.04 | 0.02 | 0.10 * | (0.83) | | | | |
| 8. CUOOCB | 4.16 | 0.50 | 0.04 | 0.01 | 0.09 * | −0.01 | 0.03 | 0.05 | 0.50 ** | (0.76) | | | |
| 9. OC | 4.07 | 0.55 | 0.10 * | 0.03 | 0.09 * | −0.01 | 0.03 | 0.00 | 0.76 ** | 0.56 ** | (0.81) | | |
| 10. ES | 4.14 | 0.61 | −0.07 | 0.06 | 0.08 | −0.12 ** | 0.06 | 0.06 | 0.51 ** | 0.33 ** | 0.49 ** | (0.79) | |
| 11. EB | 3.97 | 0.61 | 0.13 ** | −0.01 | 0.13 ** | 0.01 | −0.03 | −0.03 | 0.57 ** | 0.48 ** | 0.50 ** | 0.36 ** | (0.85) |

**Note:** EI: Environmental intent, CUOOCB: Customer-oriented OCB, OC: Organizational commitment, ES: Employee satisfaction, EB: Ecological behavior. Extraction method: Principal component analysis ($n = 508$). Rotation method: Varimax with Kaiser normalization. Factor loadings < 0.40 are suppressed. ** Correlation is significant at $p < 0.01$. *significant at $p < 0.05$.

The discriminant/divergent validity was calculated by following the approach of [70], which asserted that the square root of average variance extracted (AVE) of each construct should be greater than the correlations of this construct to all the other constructs. The square root of AVE in bold and diagonal elements is greater than 0.69, following the suggestion of Hair et al. (2016). All diagonal values were greater than interconstruct correlation values for EI $\sqrt{AVE}$, such that the value of EI = 0.77 is grater than the correlation of CUOOCB =0.50, OC = 0.76, ES= 0.51, and EB= 0.57. Moreover, all variables have CR and AVE greater than 0.70 and 0.50, respectively, thus fulfilling the criterion of for convergent validity. So, the criterion for both convergent as well as discriminant validity is supported (Table 4).

**Table 4.** Convergent and discriminant validity.

| | Convergent Validity | | | | Discriminant Validity | | | | |
|---|---|---|---|---|---|---|---|---|---|
| # | Constructs | CR | AVE | MSV | 1 | 2 | 3 | 4 | 5 |
| 1 | EI | 0.78 | 0.59 | 0.35 | **0.77** | | | | |
| 2 | CUOOCB | 0.79 | 63 | 0.38 | 0.50 ** | **0.79** | | | |
| 3 | OC | 0.86 | 0.51 | 0.49 | 0.76 ** | 0.56 ** | **0.71** | | |
| 4 | ES | 0.77 | 0.52 | 0.47 | 0.51 ** | 0.33 ** | 0.49 ** | **0.69** | |
| 5 | EB | 0.85 | 0.55 | 0.57 | 0.57 ** | 0.48 ** | 0.50 ** | 0.36 ** | **0.74** |

Note: EI: Environmental intent, CUOOCB: Customer-oriented OCB, OC: Organizational commitment, ES: Employee satisfaction, EB: Ecological behavior. ** Correlation is significant at $p < 0.01$ level (2-tailed). CR: Composit reliability. AVE: Average variance extracted.

## 5. Results

### 5.1. Direct Influence

The results indicate that employee EI significantly and positively influences EB ($\beta$ = 0.57 **, $p < 0.05$), ES ($\beta$ = 0.51 **, $p < 0.05$), customer-oriented OCB ($\beta$ = 0.50 **, $p < 0.05$), and OC ($\beta$ = 0.76 **, $p < 0.05$). The outcomes also explain that employee customer oriented OCB effects their OC ($\beta$ = 0.26 **, $p < 0.05$) and EB ($\beta$ = 0.29 **, $p < 0.05$) but does not influence ES ($\beta$ = 0.06, $p < 0.05$). However, there is no impact of employee customer OCB on ES. Similarly, employee OC stimulates ES ($\beta$ = 0.23 **, $p < 0.05$) and EB ($\beta$ = 0.06 **, $p < 0.05$) (Table 5).

**Table 5.** Hypothesis testing and path coefficient (direct influences).

| Structural Path | Path Coefficients | Conclusion |
|---|---|---|
| EI→EB | 0.57 ** | Supported |
| EI→ES | 0.38 ** | Supported |
| EI→CUOOCB | 0.46 ** | Supported |
| EI→OC | 0.65 ** | Supported |
| CUOOCB→OC | 0.26 ** | Supported |
| CUOOCB→ES | 0.06 (N.S) | Not Supported |
| CUOOCB→EB | 0.29 ** | Supported |
| OC→ES | 0.23 ** | Supported |
| OC→EB | 0.06 ** | Supported |

**Note:** EI: Environmental intent, CUOOCB: Customer-oriented OCB, OC: Organizational commitment, ES: Employee satisfaction, EB: Ecological behavior; ** Correlation is significant at $p < 0.05$ level(2-tailed). N.S: Not significant.

### 5.2. Indirect Influence

The results indicate an indirect influence of EI on EB ($\beta$ = 0.17 ***, $p < 0.05$) and ES ($\beta$ = 0.14 ***, $p < 0.05$) through customer-oriented OCB and OC, which is positive, thus supporting H1 and H5. The indirect impact of employee EI on ES ($\beta$ = 0.00, $p < 0.05$) through customer-oriented OCB is insignificant, thus H4 is rejected, whereas an indirect

influence of EI on EB ($\beta$ = 0.15 **, $p$ < 0.05) through OC is significant, thus supporting H2 (Table 6). The results of the mediating impacts of employee EI on ES and EB through customer-oriented OCB and OC partially agree with the stance of previous research [7].

**Table 6.** Hypothesis testing and path coefficient (indirect influences).

| Hypothesis | Structural Path | Path Coefficient | Conclusion |
|---|---|---|---|
| H1 | EI→CUOOCB→EB | 0.17 *** | Supported |
| H2 | EI→OC→EB | 0.15 ** | Supported |
| H4 | EI→CUOOCB→ES | 0.00 | Not Supported |
| H5 | EI→OC→ES | 0.14 *** | Supported |

**Note:** EI: Environmental intent, CUOOCB: Customer-oriented OCB, OC: Organizational commitment, ES: Employee satisfaction, EB: Ecological behavior, ** Correlation is significant at $p$ < 0.05 level, ***indirect impact is significant at $p$ < 0.01 level.

### 5.3. Interactive Influence

Sequential mediation analysis is applied to estimate the collective and individual contribution of each mediator, grounded in a causal order, while handling complex in-between relationships [71]. The results of sequential (interactive) mediation analysis of EI on EB ($\beta$ = 0.08 ***, $p$ < 0.05) and ES ($\beta$ = 0.07 ***, $p$< 0.05) through customer-oriented OCB and OC are significantly positive, thus supporting H3 and H6, respectively (Table 7). The positive association between employee EI and EB ($\beta$ = 0.57 **, $p$ < 0.05) had already been found to be positive in this study (Table 5). The sequential mediation findings support the stance of previous studies [4,7] (Table 7).

**Table 7.** Hypothesis testing and path coefficient (interactive influences).

| Hypothesis | Structural Path | Path Coefficient | Conclusion |
|---|---|---|---|
| H3 | EI→CUOOCB→OC→EB | 0.08 *** | Supported |
| H6 | EI→CUOOCB→OC→ES | 0.07 *** | Supported |

**Note:** EI: Environmental intent, CUOOCB: Customer-oriented OCB, OC: Organizational commitment, ES: Employee satisfaction, EB: Ecological behavior; interactive effect is significant at 0.05 level. *** indirect impact is significant at $p$ < 0.01 level.

## 6. Discussion and Conclusions

This research, based on the recommendations of previous studies and applying the interventions of TPB, investigated how employee environmental attitudes towards implementing green practices interact with employee job attitudes and behaviors to forecast their ecological green behavior in the hospitality sector. Therefore, the indirect impact of employee customer-oriented OCB and OC was empirically tested to verify our hypothesized statements. Our study results partially validated and supported the stance of previous studies. The findings support our stance that employees' organizational commitment and their customer-oriented discretionary behavior in an individual and interactive capacity magnifies the impact of environmental intent (willingness) on ecological behavior for resolving environmental problems. The results also show that employees' organizational commitment also relates to their contentment with the organization which is a significant factor for greening the hospitality sector. However, interestingly, employee customer-oriented discretionary behavior in the absence of organizational commitment falls short in developing ecological green behavior, which contradicts our suppositions in the introduction section.

### 6.1. Theoretical Contribution

The present study extends the boundaries of the study of EB and employee job attitudes in the context of implementing green practices in the hospitality sector, and provides a framework for understanding how EB and ES are stimulated by employee EI. The results of this study add to the literature of the organizational commitment and

customer-oriented discretionary behavior of employees of the hospitality sector. It develops our understanding how positive emotions (i.e., employee customer-oriented OCB and OC) promote EB by engendering a positive attitude towards customers and satisfying employees in their work and life conditions. Furthermore, this study applied the assumptions of TPB as a theoretical foundation for assessing the impact of environmental attitudes on EB and ES through employee customer-oriented OCB and OC. It envisions the significance of TPB interventions for greening the hospitality sector. The results show that customer-oriented emotions enhance employee cognitive and behavioral approaches to promote the development of OC. These findings explore the association of employee environmental attitudes and employee job attitudes to implement green practices in the hospitality sector.

### 6.2. Practical Contribution

This research has numerous inferences for executives and practitioners of the hospitality sector looking to encourage and sustain the EB and contentment of their employees. First, hotel employees may be aware of the importance of green practices, but they do not often practice EB. Therefore, organizations should arrange effective training programs for employees to promote their ecological behavior. Organizations should also contemplate making substantial investments in human capital with the intention of improving employee discretionary behavior, commitment, and level of satisfaction rather than a greater emphasis on increasing their productivity. Employee environmental attitudes are linked to employee job attitudes and EB, therefore, organizations should make positive efforts to promote employee environmental awareness, concern, and knowledge to implement green practices. Second, human resource managers should carefully recruit individuals who are highly environmentally aware and show discretionary behavior and are enthusiastic, committed, and display a global positive attitude, even in hostile and traumatic conditions. We recommend that managers focus on fostering employee environmental attitudes by encouraging them to behave discretionally and to show commitment even when they encounter negative situations. Third, to improve social working and to form strong relations between employees, managers can support relationship enhancement by managing social events. The findings highlight the positive effect of employee job attitudes on employee EB and job satisfaction. To promote this culture, supervisors, on a regulatory basis, might monitor this, aiming at regulating employee attitudes when employees interact with customers in rush hours, and start giving greater preference to other orientation over self-orientation so their helping behavior starts distracting their concentration from individual growth, job duties, and objective attainment towards others' goals. Fourth, the study results of the demographic variables suggest for managers that employee gender also has a significant positive impact on their ecological green behavior.

### 6.3. Research Limitations and Future Directions

Every study has limitations; this investigation also suffers from limitations pertinent to the sampling, the research tool, and the data analysis techniques. First, the research sample was taken from the hospitality sector of one province of Pakistan and future studies could be carried out by collecting data from other provinces, mainly from northern areas of Khyber Pakhtunkhwa (KPK) province, wherein the tourism and hospitality sector has a prominent footprint. Second, Pakistan is a developing country, therefore, the results of research in a developed country may be different due to their economies and infrastructure [72]. In future, a cross cultural research study, particularly among developing and developed countries, could give insight to see the difference related to the predictive EB of employees of hospitality sectors based in different cultures and economies.

Third, this study used a self-reporting survey questionnaire for data collection to measure the variables based on the perception of employees serving in the hotel industry. The other data collection techniques, including direct observations or conducting structured interviews, or even a longitudinal study, could have been better choices. These may expand or collapse the connection between the study variables. Future research could adopt other

research methodologies and techniques, including qualitative or mixed methodology to predict the ES and EB. Fourth, this research is conducted in a non-Western context (i.e., Pakistan). Therefore, this could create a generalization problem since the working context and culture in Pakistan are distinct from Western countries [72]. Fifth, the majority of the respondents of this study were male employees (i.e., 77.8%) of the hospitality sector, which may raise concerns about the generalizability of the findings for both genders. A future study could empirically test a predominantly female sample. Sixth, we controlled employee age, gender, marital status, educational level, job type, and work experience in order to avoid confounding effects on the observed relationships. A future study may add these different demographics of employees to observe the impact on the prediction of employee EB towards environmental problems. We only concentrated on employee EI but there is a room for other factors, including environmental awareness, knowledge, and concern that can affect employee EB in the presence of employee job attitudes.

*6.4. Conclusions*

We determined that the employees' willingness to execute environmental green practices is promoted by employee attitudes toward EB, employee customer-oriented discretionary behavior, and OC.

The study findings demonstrate that employees with positive environmental attitudes towards the customer-oriented OCB and OC possibly engage in EB in the workplace. The variances between the study results and previous studies may echo the varying factors of EB. The TPB model has allowed us to provide valuable understandings for managers to design strategies reflecting HR contributions. However, only partial information on the process is available. Serious environmental issues are critical factors for environmentally friendly enterprises. The future research in this area may facilitate managers to understand how to promote employee happiness and the overall well-being for developing ecological green behavior.

**Author Contributions:** Conceptualization, M.A. and G.A.; Data curation, M.A., L.A., and M.M.K.; Formal analysis, J.A. and L.A.; Funding acquisition, M.A., G.A., J.A., L.A., and M.M.K.; Investigation, M.A., G.A., L.A., and M.M.K.; Methodology, M.A. and G.A.; Project administration, G.A.; Resources, M.A.; Software, J.A. and L.A.; Supervision, G.A.; Validation, J.A.; Writing—original draft, M.A. and G.A.; Writing—review and editing, G.A., J.A., L.A., and M.M.K. All authors have read and agreed to the published version of the manuscript.

**Funding:** This research received no external funding.

**Institutional Review Board Statement:** All procedures performed in study ensured that human participants' involvement in the research was in accordance with the ethical standards of the institution and/or national research committee and with the 1975 Declaration of Helsinki and its later amendments or comparable ethical standards. The protocol was approved by the Ethics Committee of the School of Business Administration & Economics, National College of Business Administration & Economics, Lahore, Pakistan.

**Informed Consent Statement:** Informed consent was obtained from all individual participants included in this study.

**Data Availability Statement:** The datasets generated during and/or analyzed during the current study are available from the corresponding author on reasonable request.

**Acknowledgments:** This piece of work would not have been promising without the guidance and help of several individuals who, in one way or another have contributed and extended their valuable assistance for the success of this study. We are thankful to Alia Ahmed, Dean, School of Business Administration, National College of Business Administration and Economics, Lahore, for assistance and comments that greatly improved the manuscript.

**Conflicts of Interest:** The authors declare no conflict of interest.

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
