# Peer review of "Impact of Employee Job Attitudes on Ecological Green Behavior in Hospitality Sector"

_2199-8531, doi:10.3390/joitmc7010031_

Round 1

Reviewer 1 Report

Dear Author(s),

This research is of interest in the current conditions. Starting from the theory of planned behavior (TPB), the research develops and examines the direct and interactive effects of environmental attitudes and behaviors of employees on ecological practices. The authors used a sample of 508 people in the hotel industry. However, the work needs several improvements, as follows:

  1. The introduction of the section should emphasize the need for this research. The same section must present several studies and results obtained to correctly frame the research.
  2. The results section should highlight the differences obtained compared to the studies in the introductory section.
  3. Conclusions section must present in a sustained way the limitations and future directions.

Best regards,

Reviewer

Reviewer 2 Report

The purpose of this paper is to examine direct and interactive effects of employee environmental job attitudes and behaviors on ecological practices using PROCESS Macros on an actual convenient sample of 508 employees working in hospitality industry. The topic of this paper is interesting and the manuscript is well written. However, the paper has to be improved in order to be ready for the publication.

The main strengths of this paper are the following:

  • The title accurately reflects the content of this study.
  • The tables and figures are presented clearly.
  • The abstract is well organized.
  • The methods employed appropriate.

First of all, the abstract of the paper is complete and stand-alone. The author(s) mentioned the contribution as well as the practical implication of this research. Furthermore, the author(s) highlighted the need and the research gap in order to conduct this survey and study this research field. Also, some details about the methodology of the paper are provided.

The Introduction is not focused. The author(s) should use the traditional structure, just 4 paragraphs: motivation, gap, method, results, and contributions. The author(s) did not present the motivation of their paper, and did not discuss about the theoretical and practical contribution of this paper. The flow of this section is not clear.

The paper demonstrated an adequate understanding of the relevant literature in the field and the author(s) cited an appropriate range of literature sources. Several concepts are presented and the author(s) discussed sufficiently their interrelations. Furthermore, hypotheses are clearly identified and analyzed in the theoretical background section.

The research on which the paper is based is well designed and the methods that have been employed are appropriate. The author(s) presented the results of the analysis, discussed the main findings of the study and made a comparison with findings of previous papers. More information about what are the main findings, what are the main challenges/ problems as well as the opportunities should be added. Then, a comparison between the results of this study and the findings of previous studies would be interesting. Finally, the author(s) provided limitations and suggestions for future research. However, the author(s) should answer the following questions in order to make the contribution of the paper explicit in conclusion:           

  • What does this research tell us that we didn’t already know?
  • What is the contribution of the most significant results of the paper?

The flow of conclusion is not clear. The author(s) should use the following structure: main findings, theoretical contribution, practical contribution, limitations and suggestions for future research. Finally, references should be removed from the paragraphs presenting the theoretical and practical contribution.

The author(s) should update references.

Author Response

Pleas see the attachment

Reviewer 3 Report

The paper approaches an interesting subject in an interesting manner.

Some language adjustments are necessary in order to make the paper publishable.

Also, I suggest using the same style for references in the paper.

Round 2

Reviewer 1 Report

Dear Authors,

All my recommendations was implemented. 

Best regards,

Reviewer.